

# A framework for cut-over management

Guido Nageldinger

Department of Testing and Release Management, Otto (GmbH & Co KG)—A member of the
otto group, Hamburg, Germany

## ABSTRACT

The purpose of this paper is to provide a governance structure for IT-related projects in order to assure a safeguarded and timely transition to a productive environment. This transitioning, which rarely exceeds a weekend, is colloquially called 'cut-over', 'rollout' or 'deployment'. The governance structure is defined in accordance with a set of project-specific deliverables for a cascade-type procedural project-management model, which is integrated within an Information Technology Infrastructure Library (ITIL)-orientated service organization. This integration is illustrated by the use of a semi-agile release model. Due to the release model selected, which is particularly characterized by its bundling of projects for a release-specific rollout (as it is referred to in the project documentation), a new definition and interpretation of deployment from a generic ITIL perspective is required. The facilitated release model requires a distinction between a *project-specific cut-over* and a *release-specific rollout*. This separation gives rise to two types of go-live scenarios: one for each participating project and one for each release. Additionally, an interplay between cut-over planning for a project and rollout planning for a release becomes apparent. Projects should already incorporate cut-over related deliverables in the initial planning phase. Even though consulting methodologies such as ASAP (Accelerated SAP), recommend scattered, project-specific deliverables useful for cut-over planning, this publication offers an integrated approach on how to prepare systematically for a project-specific cut-over with all required deliverables. The framework provided maps out ITIL's release and deployment process by means of IT projects; furthermore it allows IT projects to interface easily with the ITIL change-management process.

## INTRODUCTION

Most IT-related projects, in particular implementation and software development projects, change a productive system landscape when taken live. On the one hand these projects face the challenge of delivering a change (within an ITIL context) in a timely and cost-effective manner. On the other hand, IT organizations need to assure the integrity of the system landscape and are required to minimize potential interference to the ongoing business, as Service Level Agreements (SLAs) need to be fulfilled.

The Central Computing and Telecommunications Agency (CCTA), a government agency in Great Britain, already developed the Information Technology Infrastructure

Corresponding author
Guido Nageldinger,
guido.nageldinger@ottogroup.com

Library (ITIL) at the end of the 1980s. The latest update of ITIL V3 was published in 2011 which is frequently referenced as ITIL 2011. Is it still relevant? Yes, because it helps bring order to IT chaos. *Proehl et al. (2013)* analyzed 255 articles in the field of IT Service Management. They also confirmed the work by *Shahsavarani & Ji (2011)*. Both studies found a growing number of published papers dealing with the development of concepts, constructs, models, methods and implementations for theory development. Performance issues in IT Service Management, justifications, and IT Infrastructure Library topics are among the most popular topics of research. Only a few articles have used or developed theories. *Marrone et al. (2014)* found organizations adopting ITIL implemented more operational level processes than the tactical/strategic level processes. The study utilizes 623 survey responses from the UK, USA, DACH (German-speaking countries) and Australia.

ITIL is rarely seen in isolation and current research focuses on the integration with IT governance standards, such as ISO/IEC 38500 and other methodologies, such as COBIT, PRINCE2 and the ISO2700x-family (*Tanovic & Orucevic , 2012*; *Shivashankarappa et al., 2012*; *Jäntti et al., 2014*; *Larsson, Rusu & Aasi, 2015*).

ITIL's release and deployment management processes are demanding. *Jokela & Jäntti (2012)* as well as *Lahtela & Jaentti (2011)* identified within their case studies the following common challenges:

- no existing process for product portfolio release and deployment management
- lack of communication
- unclear release and deployment management and/or product portfolio management process roles
- lack of resources and time for product portfolio integration, testing and reviewing
- existing models and practices are not suitable for agile software development
- uncertainty about of the contents of release packages
- high release distribution frequency
- different and tailored release packages
- the change management is not up to date and
- no existing service catalog.

The project-specific cut-over, as defined in more detail below, requires detailed planning, many meetings and several agreements with all stakeholders involved. Which questions need to be addressed? Here are just some of them:

- How should the project outcomes be transferred to operation?
- Which scenario provides the best compromise between time, cost and risk?
- What happens if the cut-over fails?
- How can the previous system condition or version be reinstalled in case cut-over fails?
- How can we ensure that projects start early enough, with all the necessary preparation work?

- How can cut-over activities be aligned within an IT service organization?
- How do we measure the success of the cut-over–and how can we maximise it?

Unfortunately, answers and activities associated with these cut-over related topics are frequently addressed too late, probably because the cut-over is one of the final steps in projects and can therefore be hidden for a long time. If a cut-over and its associated deliverables are not integrated within the overall Project Plan then projects are at risk of being delivered too late. For this reason, there is an urgent business need to govern projects in such a manner that they do not forget to cover cut-over related items. So, which items should projects cover?

This question and many others are addressed by a 'framework for deployment management' as referred to below in this document. The purpose of this paper is to provide a methodology which assures a robust and timely go-live for projects.

An itemised objective overview:

- to list the key deliverables projects need to provide in order to assure a timely and safe go-live
- to illustrate how these deliverables can be integrated within the ITIL release and deployment process
- to look into different deployment types, which this paper refers to as (a) project-specific cut-over, (b) application-specific cut-over and (c) release-specific rollout along with the timeframes and timepoints involved, and lastly
- to present a framework which enables an IT-service organization to transition a project to a productive environment.

The scope of the deployment framework introduced is required

- to be applicable within an Information Technology Infrastructure Library (ITIL)-orientated service organization, and
- to extend commonly facilitated project-management methodologies.

As with most management practices, there are always more ways to achieve similar results with other approaches. However, in order to examine and establish good management practice, these methodologies need to be published to facilitate comparison. The framework published here is just one of many potential options. It illustrates how deployment activities can be harmonised with commonly facilitated IT projects as well as quality-management practices, and integrated in an ITIL environment.

Even though it is possible to take advantage of the deployment framework presented here without ITIL, many companies have adopted ITIL recommendations and best practices and are organized in a service-orientated manner. IT projects are therefore unlikely to exist in isolation and are more likely to be embedded in a service-orientated manner. ITIL is known to be generic and by itself is not complete. Its purpose is not to provide advice on how to implement IT Service Management processes. Instead it requires integration with other disciplines such as project management and quality management.

The framework presented here in the form of an illustrative case study arose during my consultancy work and has been implemented for some applications within the Department of Testing and Release Management of Otto (GmbH & Co KG) which is a member of the otto group. It is not a hypothetical model. Projects which wish to cut-over in form of a release-specific rollout are requested to comply with this framework. Within every release between 15 to 25 projects are participating. Six release-specific rollouts are conducted every year. Even though consulting methodologies such as ASAP (Accelerated SAP) recommend scattered project-specific deliverables useful for Cut-over Planning, this publication offers an integrated approach on how to systematically prepare for a project-specific cut-over with all required deliverables. The framework provided extends the ITIL release and deployment process by means of IT projects and allows IT projects to interface easily with the ITIL change-management process.

The activities of the otto group, with its headquarters located in Hamburg, Germany, are grouped into three main business areas: (i) Multichannel Retail, (ii) Financial Services and (iii) Service. This structure is consistently reflected in the group's activities along the retail value chain, in Logistics for instance. In the year 2014/15, the Group generated consolidated revenue of more than 12 billion euros and employed about 54,000 staff worldwide.

## PROJECT-SPECIFIC CUT-OVER VERSUS RELEASE-SPECIFIC ROLLOUT

ITIL is often facilitated as a checklist. With regard to the release and deployment process, it consists of the following steps (*Rance, 2011*):

- plan and prepare a release
- build and test a release
- plan and prepare deployment
- conduct deployment
- conduct review, and
- provide early-life support.

Due to its generic nature and the need to interface with other disciplines, IT-service organizations implement these steps quite differently. The variability of implementation options may be caused by different interpretations of the term *'release'*, which is also related to the term *'change'*. How does ITIL define these terms?

The term 'release' is defined as: *"One or more changes to an IT service that are built, tested and deployed together. A single release may include changes to hardware, software, documentation, processes and other components."* (*AXELOS, 2011*).

The term 'change' is defined as: *"The addition, modification or removal of anything that could have an effect on IT services. The scope should include changes to all architectures, processes, tools, metrics and documentation, as well as changes to IT services and other configuration items."* (*AXELOS, 2011*).

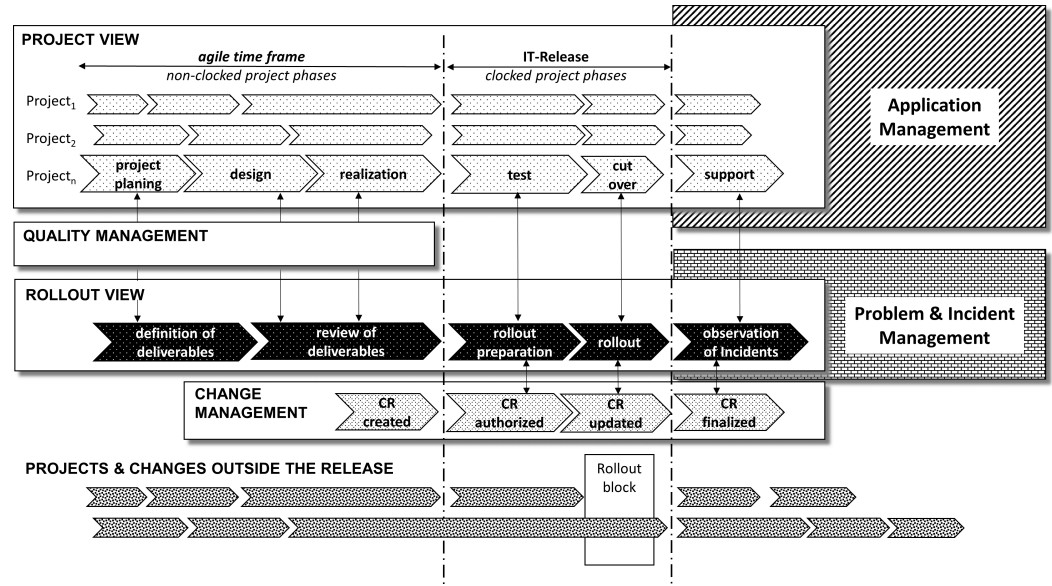

**Figure 1** **Example of a semi-agile release model.** Example of a semi-agile release model with its project and rollout view and relation to ITIL's Application Management, Problem & Incident Management as well as Change Management.

In order to safeguard the related system landscape as well as associated services, conventions need to be outlined on the practical arrangement and organization of release-related changes. Here, the term '**release-model**' is facilitated for a set of rules and conventions used to bundle and organize a release.

Figure 1 illustrates a semi-agile release model, which is used within this publication as an example to address some dependencies (*Nageldinger, 2015*). This release model is called 'semi-agile' here because it consists of an agile section and a release phase, which follows a classical sequential pattern (cascade model). During the agile period, phases between projects are non-clocked. Project phases which fall within the agile period relate to the Project Planning, design and realization section, and can be conducted in sprints. Once these projects participate in an integration test, then their phases need to be clocked. Independently of which release model is facilitated, the release phase will most likely consist of (i) an entry gate, (ii) technical and business-related integration tests with their acceptance procedures, and (iii) the release-specific rollout (see Fig. 2).

Let us now look at the deployment types. In case the bundling of projects is foreseen by the release model, such as in the one presented here, then we encounter deployment related to the release as well as deployment related to participating projects. ITIL does not distinguish between these. Here, the term '*deployment*' (*AXELOS, 2011*) is defined as *"an activity responsible for movement of new or changed hardware, software, documentation, process etc. to the live environment. Deployment is part of the release and deployment management process."* It can probably be argued that such a distinction is unnecessary, since all bundled projects are to be deployed in the same time slot. However, the project-specific deployment and its associated preparation work are owned by the

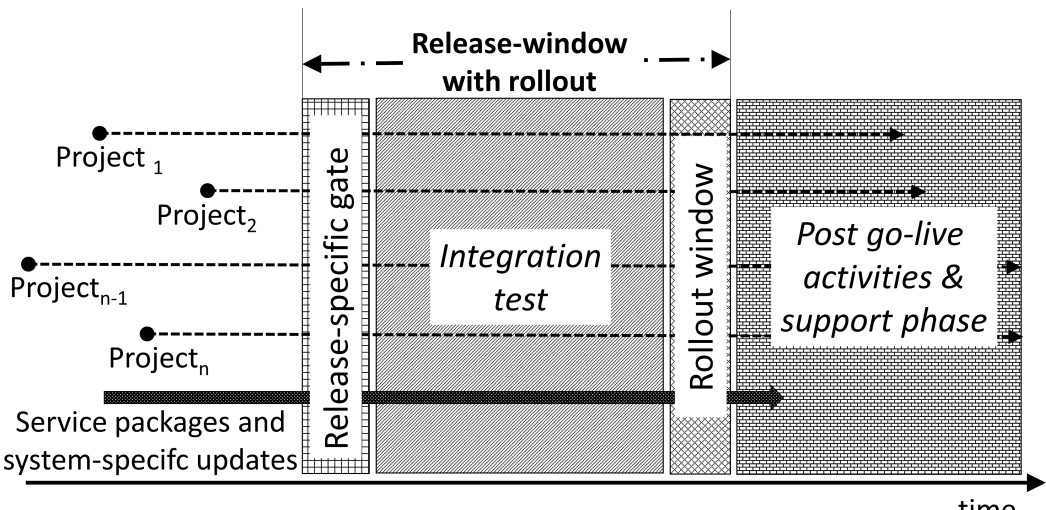

**Figure 2  Release phase and rollout window.** Illustration of the release phase and the positioning of the rollout window.

Project Manager and, in the case of larger projects, by a designated Cut-over Manager. According to ITIL the Release and Deployment Manager owns the release and its associated deployment.

ITIL's definition of '**deployment**' lacks conceptual clarity. This is because if we look only at the movement of software to the live environment then we also need to distinguish between the actual movement and its use when the software is 'switched on'. This is necessary if legal changes come into effect at a specific point in time, for example, but the software is required to be moved to the live environment beforehand. For the purposes of this paper the term '**deployment**' from now on is solely used to refer to the movement of software to the live environment, which is commonly conducted within a restricted timeframe. The term '**point-of-use**' is facilitated to account for the point in time in which the software is 'switched on'.

In order to safeguard our system landscape and associated services we need to monitor both the deployment (timeframe) and point(s)-of-use (point in time). In the context of a release, one deployment and potentially more points-of-use are scheduled. This aspect is elaborated further below in the discussion of the release-change.

Besides the participating projects, a release additionally includes service packs (a phrase used for minor upgrades), bug fixes and smaller system-related features. These non-project elements are usually governed by separate changes. All ingredients of a release should provide the option of independent transitioning to the productive environment. If this option is not provided, then projects or non-project elements cannot easily be excluded from the release if they fail the integration test. Smaller projects are likely to be bundled in form of a release, and then transitioned to a productive environment in form of a release-specific rollout. Major IT implementation projects are preferably transitioned to a productive environment independently, in order to reduce complexity.

The term '**project-specific cut-over**' defines the transition of a project to a productive environment. It relates to the project as a whole and includes the software deployment and its point of use. The actual timeframe associated with the project-specific cut-over is called the '**cut-over window**'. The '**release-specific rollout**' defines the collective transition of bundled projects to a productive environment. In the same way, the associated timeframe is called the '**Rollout Window**' (*Nageldinger, 2014*).

Frequently, projects communicate what is frequently called a 'go-live date' on their overall project chart. In view of the reasons given above, this could mean (i) end of the software deployment, (ii) end of cut-over window, which relates to the transitioning of the whole project, or (iii) point-of-use. It is therefore recommended to question what exactly this date refers to.

Beside the project-specific cut-over and the release-specific rollout we are most likely to face a third transitioning aspect, here called the '**application-specific cut-over**': this relates to a specific system or application and mainly covers approaches related to maintenance or upgrades. It differs from the project-specific cut-over in that it consists of canonical (i.e., reoccurring with every rollout) activities and tasks. The deployment of service packs can be organized canonically, for example.

Within an SAP context the term 'rollout' is frequently associated with a so-called 'template approach'. In this, the core configuration is embedded within a reusable template and the country-specific characteristics are added separately in the form of a local configuration (*SAP, 2009*). SAP's usage of the term 'rollout' is quite similar to the definition provided here and its associated release context. This is because the release-specific rollout also consists of reoccurring activities, here referred to as "application-specific cut-over", which is similar to SAP's template approach. The project-specific cut-over activities, which are non-reoccurring, may be considered in the same way as SAP's country-specific characteristics.

The **Cut-over Schedule** is a list of tasks and activities required in order to conduct the cut-over. It differs from the ordinary overall project schedule as it only focuses on tasks and activities within the cut-over window, of short duration and lasting only for a couple of hours or a weekend. The Cut-over Schedule is created by duration-based scheduling techniques. Additionally, it contains a list of prerequisites, such as the delivery of training courses and the installation of printers etc. These prerequisites need to be completed before the actual cut-over can be conducted and are part of the overall project schedule. The Cut-over Schedule does not relate to the post go-live phase. For this objective projects need to have a separate plan, which for example addresses aspects of support, data-conservation and accounting closure procedures.

Besides the Cut-over Schedule the Cut-over Plan contains a variety of other deliverables, in the same way as a project-management plan, which are further elaborated below. Key elements of a Cut-over Schedule are for example termination points, frequently called '**points-of-return**' (POR). Theses PORs aim to trigger activities used (i) to restore the initial condition or (ii) incrementally fall back to a previous POR. The last POR is referred to as the '**point-of-no-return**' (PONR). The PONR does not always fall within the cut-over

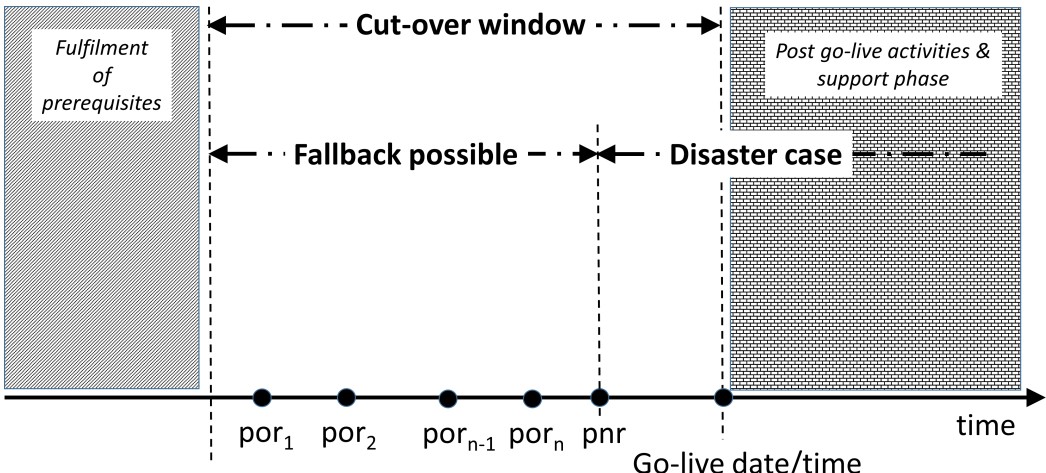

**Figure 3** Visualization of the cut-over window.

window. The Fallback Concept is the name of a document used to describe activities which are potentially necessary at these PORs, in case the system needs to be rolled back to a previous state or even its initial state. If the cut-over activities pass the PONR then we enter the disaster scenario. For such events, a disaster-recovery concept defining appropriate measures should be in place. The disaster-recovery concept is mentioned here because it is unfortunately frequently neglected. It is regularly associated with earthquakes, hurricanes or terrorist attacks. However, unforeseen events during a cut-over after the PONR can also trigger the disaster scenario (*BSI, 2008*). The cut-over window is illustrated in Fig. 3.

The sections above clarify why certain release-models, which foresee the bundling of projects for collective rollout, require at least two perspectives—a project-specific and a release-specific one. An important outcome for the preparation of a project-specific cut-over and a release-specific rollout is the go-live scenario. A go-live scenario defines the overall approach on how a cut-over or rollout is to be conducted. The go-live scenarios differ in most cases, which is illustrated by an example in Fig. 4. The go-live scenario of project A utilizes the Rollout Window of release 1 and 2, whereas projects B and C merely take advantage of one Rollout Window. The release-specific rollout also facilitates go-live scenarios. Here, the rollout of release 1 uses a sequential rollout strategy, while the rollout of release 2 uses a parallel rollout strategy.

## RECOMMENDED DELIVERABLES AND OUTCOMES FOR THE PREPARATION OF A PROJECT-SPECIFIC CUT-OVER

This chapter covers recommended deliverables and outcomes for projects in order to prepare for a project specific cut-over. Projects are assumed to be arranged in a cascading manner with classic phases, in order to keep this publication as simple as possible. However, as stated above, activities may be undertaken in an agile manner before entering the release phase.

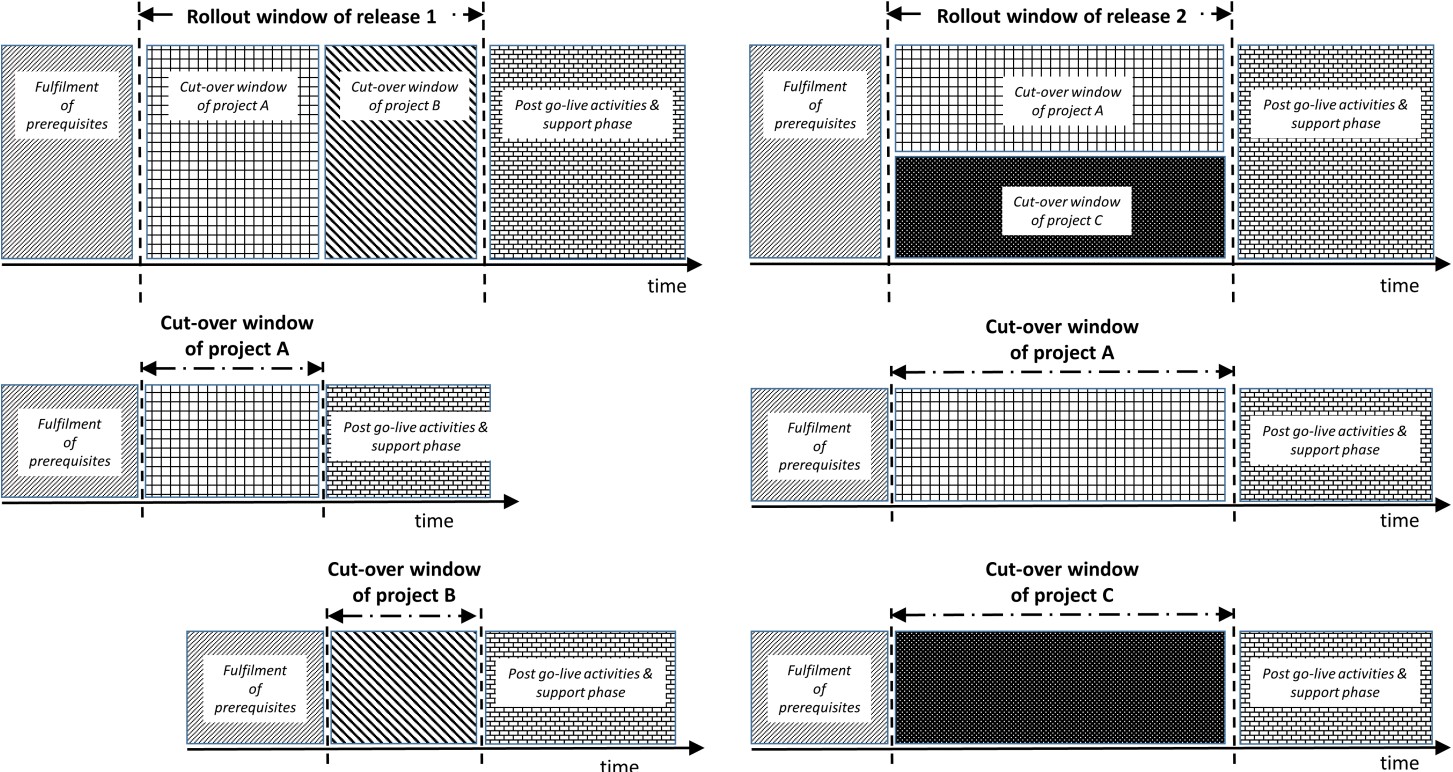

**Figure 4 Illustration of two release-specific rollouts.** The Go-live Strategy of project A utilizes the Rollout Window of release 1 and 2, whereas projects B and C facilitate only one Rollout Window. In the same way as the project's Go-live Strategy, the release-specific rollout uses a Go-live Strategy for the release. In this illustration, the rollout of release 1 utilizes a sequential rollout strategy and the rollout of release 2 a parallel rollout strategy.

The release model presented in Fig. 1 illustrates the following generic phases, which are quite similar in the case of IT-related projects:

- Project-planning phase
- Design phase
- Realization phase
- Test phase
- Cut-over phase
- Support phase.

Quality Management is facilitated in order to request these cut-over relevant project deliverables, which are then reviewed by Rollout Management. Table 1 illustrates potential deliverables (*Nageldinger, 2015*). Some of them are mandatory, others are optional. These deliverables are further elaborated here below and presented according to their project phases.

**Table 1 Project-specific deliverables for cut-over planning.** Bold font used for deliverables related to Rollout Management; italic font used for optional deliverables which are not evaluated by Quality Management; normal font used for mandatory deliverables.

| Project planning | Design | Realization | Test | Cut-over | Support |
|---|---|---|---|---|---|
| Plan for Deployment Management | *System-profile description* | *Decision model presentation for the Go-live & Fallback Strategy* | FINAL Cut-over Plan | Operational Cut-over | Observation of incidents |
| Establish Cut-over Manager (COM) position | System landscape design (SLDe) | Decision related to the Go-live & Fallback Strategy | Fallback concept | | **Lessons learned** |
| **Initial dialogue with Rollout Management** | Extension to SLDe due to Go-live Strategy | One-Pager | Disaster- recovery concept | | |
| | Go-live strategy | DRAFT Cut-over Plan | **Plan for the deployment of personnel** | | |
| | Fallback Strategy | Risk Register | Cut-over test | | |
| | *Framework for cut-over planning* | Dependency Matrix | *Go-live simulation* | | |
| | *Stakeholder analysis* | **Dialogue with Rollout Management** | | | |
| | *Kick-off* | | | | |
| | *Communication Plan* | | | | |
| | *Completely defined Cut-over Team* | | | | |

**Table 2 Activity types used for a Rollout/Cut-over Schedule.** Ordinary activities are not assigned an activity type.

| Activity type | Explanation |
|---|---|
| Info | Someone needs to be informed, usually by e-mail. |
| Checkpoint | Technical verification point within cut-over/rollout flow. |
| Optional | Activity is conducted only for certain rollouts; if the activity is not required then the time is set to 0. |
| Security | Security-related activity, such as back-up or the implementation of restore points. |
| Handover | Handover of a certain document, protocol etc. |
| Unique | Non-recurring activity within the rollout flow; commonly relates to a project and is therefore not facilitated for other rollouts. |
| Breakpoints | Breakpoints are used during task or process chains in order to verify intermediate results. This activity type relates to these breakpoints. |
| Comment | Project-planning tools often provide limited functionality related to comments within the working breakdown structure (WBS). Comments are activities with time = 0 and are merely used to provide additional information within the WBS. |

## Project-planning phase

The preparation of a project-specific cut-over already needs to be considered during the planning phase of a project. Key deliverables are (i) a decision related to the establishment of a Cut-over Manager position, (ii) an agreement on cut-over specific deliverables, which are included (iii) within a plan for Deployment Management.

### Plan for Deployment Management

In order to deliver a Cut-over Plan on time it is important that the cut-over related deliverables and outcomes are already taken into consideration during the planning phase of a project. Larger projects will probably require a separate schedule for all cut-over related planning tasks and activities, whereas smaller projects include them in their overall project schedule. Here, this key deliverable is called deployment management plan. Please note that these tasks relate solely to the necessary preparation and planning work and not to the Cut-over Schedule itself.

### Establish Cut-over Manager (COM) position

During the planning phase it needs to be decided whether a Cut-over Manager (COM) position is to be implemented. The COM role may be compared with the role of a Project Manager; it is solely an administrative role which covers IT-related topics as well as business related ones. The only difference is that the COM focuses on the preparation work associated with activities conducted within the cut-over window. A COM is responsible for the preparation, planning and execution of the project-specific cut-over. He or she needs to be positioned beside the Project Manager and should be part of the Steering Committee. This is because conflicts of interest are likely to occur between the COM and the Project Manager. Additionally, a COM needs to drive decisions and potential change-requests related to the cut-over; many of these bear associated risks. A common role for a COM would include the following responsibilities:

- drive a decision on the final go-live strategy
- implementation and guidance of a Cut-over Team
- creation of a Cut-over Plan
- creation of a list with initial conditions required to be met prior to the cut-over
- initiating a list of activities after cut-over
- planning, execution and organization of the cut-over-test and go-live-simulation.

Smaller projects do not usually need a COM as the Project Manager fulfils this role.

### Initial dialogue with Rollout Management

During the project-planning phase an initial dialogue with Rollout Management is mandatory because related deliverables need to be scaled to the project size. This scaling is quite difficult to automate since it depends on experience and the potential interfaces involved. A simple formula, for example based on the project budget, is insufficient.

## Design phase

Prior to completion of the design phase, the Go-live Strategy and Fallback Strategy should be finalized. Depending on the project size, these key deliverables can be combined and should provide a top-level description of options on how to transition the project to a productive environment, as well as how the initial state can be restored.

### Go-live strategy

The Go-live Strategy is of an investigative nature. It should provide several potential go-live options with their associated impacts, rather than an ultimate solution. It should foster discussions related to the topic. The idea is to provide a general overview and associated opinions. This document is later used to balance stakeholder interests and to carve out the final go-live method.

A Go-live Strategy starts with the go-live scope. How does the go-live scope differ from the project scope? Practically speaking, the *go-live scope* is the elaborated version of the initial project scope and it is therefore more concrete. The *project scope*, for example, can state the introduction of a system to meet the purpose XYZ. A go-live scope needs to state precisely which system is going to be implemented, lists the technology involved and states the required organizational changes. A project scope, for example, can state the improvement of a process. The go-live scope needs to state how a particular process is to be improved and which changes are required to be conducted. It is recommended to define the go-live scope in writing during a workshop attended by all project key players. This exercise is valuable for the overall project, as well as in identifying scope changes already occurring. A properly defined go-live scope is a prerequisite for evaluating all other cut-over relevant questions.

The Go-live Strategy covers the following aspects:

- a description on how the project can be transitioned to the live environment. The description can be of generic nature and should relate to the major approaches, such as big-bang, phased go-live, or iterative methods.
- It is highly recommended to provide several go-live options which identify and provide for the various associated risks.
- Many go-live scenarios potentially impact the business and technical infrastructure. The impact and its potential consequences should be discussed with the stakeholders. Their opinion should be included within the Go-live Strategy and referenced as well.
- Alternative go-live approaches, which are probably not investigated further, should be defined and recorded. Additionally, it should be recorded why particular options are not followed up.
- One section should focus on potential go-live dates. Again, rather than providing one ideal date, several potential dates should be evaluated. What are the benefits and catches of these dates? Which dates could be used as an alternative, in case of project delays? Who recommends these dates? What is the reasoning behind them?
- A cut-over window usually last for just a few hours. However, some projects require a weekend. Which constraints and risks are associated with the cut-over timeframe? What are the stakeholders' wishes? Why do they request a shorter or longer timeframe?
- Many projects alter existing *business processes*. These changes should be addressed in form of an AS-IS and TO-BE description. Which processes are newly introduced, altered or deactivated? Departments which are impacted should be identified as well. Who are the key contact persons?

- Some projects alter *the organization*. If this is the case, what does the new organization look like? A section of the Project Plan should describe and illustrate the old and new organization. It should highlight the targeted changes, as well as how the targeted changes are to be achieved. Are the change constraints related to the timeframe? Projects with significant reorganization focus should employ an organizational change management task force which delivers this information.

- Since people tend to change their opinion or might need to be contacted later for this, further discussions on all related references should be logged.

- How many people are going to be impacted during and after a potential go-live? The impact can be of a technical, organizational and/or business nature.

It is an open secret that many companies do not document their business processes. Major implementation projects therefore require this task to be conducted. The Go-live Strategy needs to focus on the targeted change by comparing the AS-IS with the TO-BE state. It outlines transition options in order to cut-over a project to a productive environment. These transition options also include the technical aspects and system landscape. However, the technical documentation of the AS-IS and TO-BE is usually conducted in a separate document, here called a 'System Landscape Design' (SLDe) document.

### System Landscape Design (SLDe)

The creation of an SLDe is commonly the task of a core technology or IT architecture department. For further Cut-over Planning it is obviously essential that this document exists, and that the current as well as the future system landscape is not only presented in a graphical manner but also properly described. Even though some companies have a Configuration Management Database (CMDB) in place, the information available is frequently not newsworthy or lacks information required by the project. The creation of an SLDe is therefore a task which needs to be foreseen in most IT implementation projects. From the cut-over perspective additional questions need to answered, for example:

- who are the applications contact persons (such as Business Owner, Administrator, key users etc.)?
- Which business process are linked to the AS-IS and TO-BE system landscape?
- Which departments currently work with the applications involved?
- A list of associated applications
- A list of interfaces, their business objective and the technical protocol used.

### Extension to SLDe

Certain go-live approaches require an extension of the SLDe, which is treated here as a separate deliverable in order that it is not forgotten. Extensions are, for example, particular interfaces for data-migration purposes. System landscapes related to the proposed test methodology as well as the go-live simulation and the cut-over test, which are explained further below, are also important extensions of the SLDe.

### System profile description

If a project is initiated in an environment where a CMDB is missing or not sufficiently maintained, then system-profile descriptions need to be created; this is commonly conducted in the form of a survey. Such a survey should be conducted together with the department which already owns the CMDB. It is therefore advisable during the project-planning phase to conduct a brief review on how well the system landscape is documented and to foresee the system-profile descriptions as an evaluation task within the project schedule, since this task can become quite time-consuming.

### Fallback Strategy

The Fallback Strategy should be defined in writing within the same timeframe as the Go-live Strategy. It provides a top-level draft of the potential activities required in order to reinstall the initial state prior to the start of cut-over. The detailed version of this document, here called 'Fallback Concept', will be created later and finalized during the test phase. The Fallback Strategy addresses the following questions:

- how can the system landscape be rolled back to its state prior to the start of cut-over? Are further scenarios possible?
- How long would such a fallback procedure take?
- Which criteria or conditions should trigger a fallback?
- Which risks are associated with each fallback option provided? What do their mitigation strategies look like?
- In case of a fallback, how great is the potential business impact? Can the potential impact be quantified? How great is the impact in case a foreseen fallback scenario fails? How likely is it to occur? How great is the impact in case the system landscape involved is unavailable for one day? How many days can a business survive without IT?
- Which model should be used in order to calculate the business impact?
- What is the maximum acceptable downtime? How is it calculated? Are Service Level Agreements (SLAs) jeopardized? How great are the potential contractual penalties?
- Where can potential PORs be positioned? Where is the PONR? Do all relevant systems have a disaster recovery concept in place, in case the PONR has been reached and things go wrong? Is it possible to assign a potential business impact to each POR identified? What are the termination criteria associated with each POR?

The questions provided are obviously not complete, but are intended to illustrate the potential complexity involved.

### Communication-related deliverables

Cut-over Management is communication management, and related deliverables such as a Communication Plan, a stakeholder analysis and a kick-off, to mention just a few, do not differ from ordinary project-management methodologies and are therefore not further elaborated here. However, these methodologies need to be applied within the cut-over context. A warehouseman, for example, is unlikely to be part of an ordinary

project stakeholder analysis. However, within a cut-over context, a warehouseman can be an important key player. By the end of the design phase the Cut-over Team should be defined. The Cut-over Team consists of all people required to contribute to the cut-over deliverables explained within this section. The Cut-over Team also should be completely defined by the end of the design phase.

Communication-related deliverables are recommended not to be evaluated during a potential assessment. This is because both a Go-live Strategy and a Fallback Strategy require these communication items to be in place in order to prove that all important stakeholders' views have been considered.

The framework for Cut-over Planning can also been seen as a communication-related deliverable. It is a document which describes how planning for the cut-over is to be conducted, and which cut-over related deliverables are to be produced. Such a document is obviously only advisable for larger cut-over tasks and is therefore not mandatory.

## Realization phase

### Decision related to the go-live and Fallback Strategy

The decision related to the go-live and Fallback Strategy is a key outcome of the realization phase. It needs to be conducted by the Steering Committee since it relates to the triple constraints, such as scope, timeframe, budget and risk. The decision is required to be prepared in form of a decision-model presentation and commonly extensively discussed, prior to a Steering Committee meeting. It is then up to the members of the Steering Committee to consult or involve further senior management if the go-live risks associated require this. This decision may present several political challenges and if it is conducted too late then the timely delivery of the project is at risk.

### One-pager

The One-Pager summarizes the go-live and Fallback Strategy decided. It is later integrated within the release-specific Rollout Handbook. The objective of the one-pager is to inform all other projects and participating parties about the rollout and to facilitate the preparation of the release-specific rollout.

### Draft Cut-over Plan

The Cut-over Plan and the Cut-over Schedule are not the same and are explained further below, within the context of the Rollout Handbook. The Cut-over Schedule is an important part of the Cut-over Plan. As a minimum requirement a draft Cut-over Schedule should be available at the end of the realization phase, sufficient to support the planning activities for the release-specific rollout. Since the term 'draft' can be interpreted in several ways, it is suggested to define the outcome in advance in order to avoid disappointment at the end of the realization phase. This draft schedule should contain all activity blocks, with their subordinated activities as well as their durations, dependencies within these activity blocks, with an attached working breakdown structure (WBS) and a Glossary.

### Risk register

The cut-over related Risk Register does not differ much from an ordinary project Risk Register. The administrative work of this Risk Register should be kept to a minimum, since the risks solely relate to the cut-over. However, it is still necessary that such a record is properly administrated. Since key elements of such a Risk Register are quite frequently forgotten, such as the Risk Owner or Risk Indicator, a practical set of parameters are suggested here, such as:

- **Risk ID:** the ID number assigned to the risk
- **Risk Category:** a sorting criterion used to categorise risks. Commonly used categories are, for example, applications or phases during a rollout.
- **Risk Description:** this can follow a simple scheme by addressing three questions: (i) What is the actual risk? (ii) How is the risk caused? (iii) What is the impact if the risk does occur?
- **Alerter:** can be anyone who identifies and reports a risk.
- **Alerting Date:** the date on which a risk has been recorded.
- **Magnitude:** calculated on the basis of the probability and potential impact of a risk occurrence.
- **Risk Indicator:** criteria used to recognize that a risk has manifested itself; for instance, error messages after a database update etc.
- **Mitigation Measure:** a counter-action that can be taken if a risk manifests itself.
- **Risk Owner:** a person with the assigned authority and competency to mitigate a risk. The Risk Owner identifies mitigation strategies, defines Risk Indicators and nominates a Technical Contact for the respective rollout. This role also includes refining the Risk Description, since the risk as such is frequently confused with the causes of a risk.
- **Technical Contact:** a technical expert or a team which shares the required technical expertise and is in any case present during the rollout, usually fulfils this role. This person is responsible during the rollout (i) to verify whether a risk has manifested itself, and (ii) to coordinate Mitigation Measures defined within the Risk Register.
- **Processing Status:** describes how far the processing of the risk has progressed.
  - **New:** a new risk has been recorded, described and assigned to an initial Risk Owner;
  - **Under Way:** the Risk owner has accepted his role for the particular risk raised, and communicates a delivery date on which the Mitigation Measures and Risk Indicator are defined;
  - **Completed:** as soon as the Mitigation Measures and Risk Indicator are described and a Technical Contact for the particular risk has been nominated, Risk Processing has been completed.
- **Completion Date:** the date on which processing status is set to 'Completed'.

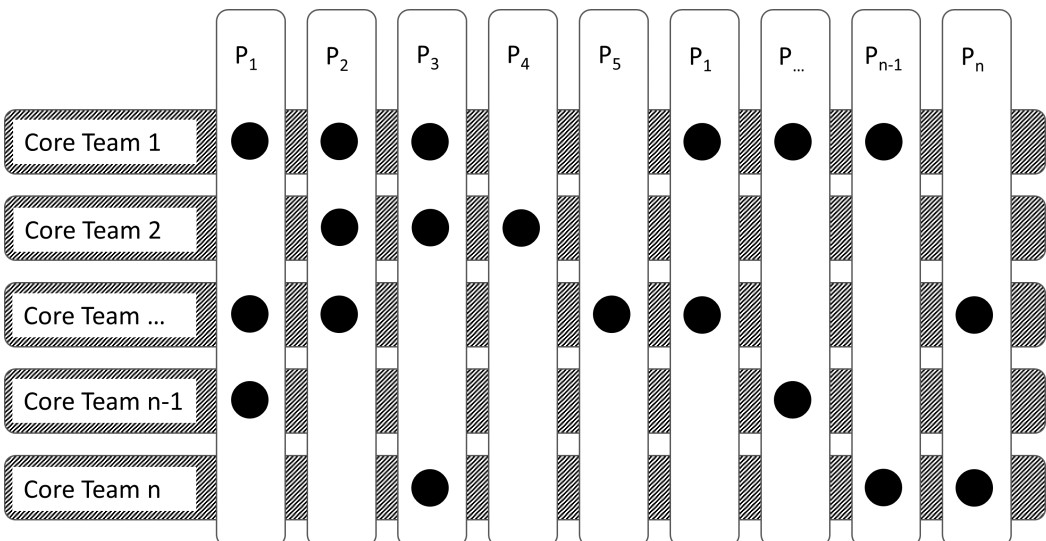

**Figure 5** **Dependency Matrix.** Illustration of a Dependency Matrix, which is used as a tool for the evaluation of the (potential) impact and interrelations between Project Teams and Core Teams; each bold point symbolizes an interaction between a Project Team and a Core Team; the sum of interactions is a criterion for the complexity of a project as well as a release.

## Dependency matrix

A collective participation by projects in a release-specific rollout causes a need to manage dependencies. These dependencies can occur between Project Teams as well as jointly used personnel resources, here called 'Core Teams'. The Dependency Matrix is a tool used to evaluate the potential impact and interrelation between Project Teams and Core Teams. Project Leads are requested to fill out a template and to identify which key personnel are required to be present during the cut-over.

The Dependency Matrix is hard to define as a tool because the terms (i) 'Core Team' as well as (ii) 'dependency' are themselves difficult to define. Here, a Core Team is seen as a heterogeneous group of key personnel required to be present during a cut-over/rollout. It consists, for example, of administrative personnel related to key applications and databases as well as Service Owners (within an ITIL context). A dependency between a Project Team and a Core Team is present if communication between the Project Team and the Core Team or its related technology is necessary during cut-over/rollout. Since Project Teams usually identify similar Core Teams, a template can be created. This is illustrated in Fig. 5. The bold points visualize dependencies between Project Teams and Core Teams participating in a release.

Even though the term 'Core Team' is treated less strictly, these dependencies can be facilitated to quantify the complexity of a release. Complex projects usually require more key personnel than less complex projects. A complex release usually consists of several projects with many dependencies. Therefore, one possible measurement of the complexity of a release is the total amount of dependencies. This idea is further elaborated below. Additionally, such a Dependency Matrix is quite a sensible instrument to have in order to

identify projects which have probably influenced a particular business process negatively after rollout.

### Dialogue with Rollout Management

Projects are requested to pass a gate prior entering the release phase, which consists of a variety of project deliverables. Most of the cut-over related deliverables have already presented here, such as (i) the decision related to the Go-live Strategy, (ii) the Cut-over Plan (draft), (iii) a Risk Register, (iv) Dependency Matrix and (v) an obligatory dialogue with Rollout Management. The cut-over related deliverables are discussed during the dialogue along with a couple of questions which the Rollout Manager is to address.

## Test phase

During the test phase the Cut-over Plan and Fallback Concept need to be finalized. The disaster recovery concept relates to the period after the Point-Of-No-Return (PONR) and is required to be produced for new systems or potentially extended for existing systems. It has already been mentioned that an insufficiently documented system landscape can severely affect a project's progress. It should therefore already been evaluated during the Project Planning phase whether sufficiently documented disaster recovery concepts are in place for existing systems.

After completing the integration tests, a cut-over test and ago-live simulation should be conducted. The cut-over test is usually the last test prior to the rollout with the objective of validating the Cut-over Schedule. Some steps are simply conducted in an exemplary fashion, such as data migration. This is in particular valid if a huge amount of data is required to be migrated. A cut-over test would also cover a couple of fallback scenarios and (if necessary) a disaster recovery case as well. A second form of validation is what this paper refers to as the 'go-live simulation', which has more similarities to a rehearsal than a test. The go-live simulation focuses on the communication requirements during a cut-over and necessary delivery points. It should be conducted with the whole Cut-over Team (personnel involved in the cut-over) in place under circumstances which are close to the real cut-over situation. For this reason the plan for the deployment of personnel should be in place as well. Both the cut-over test as well as the go-live simulation can be conducted solely for a particular project and/or together with other projects as part of a release. The scope of both validation forms needs to be defined as part of the decision related to the Go-live Strategy during the realization phase. The effort required to conduct these tests is quite significant, since IT landscapes close to the live environment are required.

## Cut-over and support phase

During the cut-over phase the operative cut-over is to be conducted either with other projects in form of a release, or by one project alone. In the latter case, the project does not participate in a release and conducts a project-specific cut-over independently. Such an independent cut-over is most likely conducted in large-scale projects.

ITIL recommends a support phase as part of the release and deployment process (*Rance, 2011*). However, support can only be provided by the personnel who have supporting

knowledge, that is, knowledge which can be provided exclusively by the Project Teams and not by the personnel owning the release and deployment process. It is therefore argued that after the end of the Rollout Window, the subsequent support phase is required to be covered by the Project Teams.

It is advisable to incorporate a specific period for the observation of incidents after cut-over or rollout prior to the Lessons-learned Workshop. This observational period lasts from 1 to 4 weeks and obviously depends on the complexity of the project or release. This aspect is further developed below, together with recommendations on how to run a Lessons-learned Workshop.

## RECOMMENDED DELIVERABLES AND OUTCOMES FOR THE PREPARATION OF A RELEASE-SPECIFIC ROLLOUT

This chapter highlights a couple of important deliverables and outcomes of the release-specific Rollout Management. Large-scale projects which are likely to cut-over independently may consider the outcomes described here as well, even though these are assigned to the release phase.

### The Rollout Handbook

The planning document and key deliverable for the release-specific rollout is the Rollout Handbook. This may be compared with the Project Plan defined in the PMBOK®. However, it relates solely to the release-specific rollout. The PMBOK® (*PMI, 2001*) defines a Project Plan as a "… formal, approved document used to guide both project execution and project control. The primary uses of the Project Plan are to document planning assumptions and decisions, facilitate communication among stakeholders, and document approved scope, cost and schedule baselines. A Project Plan may be summarized or detailed."

Even though the Rollout Handbook is structured to the same degree as a Project Plan, it is only finalised shortly prior to the actual rollout. A classic Project Plan, however, needs to be completed at the beginning of the project, during the planning phase.

The Rollout Handbook consists of the following items:

**Executive Summary:** describes the major rollout activities, participating projects and applications, highlights major risks and provides an overview chart of the Rollout Windows.

**Version History:** the Rollout Handbook is to be extended and adjusted throughout the whole release phase. It is therefore important to publish temporary versions of this book in regular iterations, so that it can be reviewed. A Version History is therefore crucial in order to keep track of these changes.

**Communication Plan:** lists all participating persons with their contact details. Communication related to the rollout is part of the Rollout Schedule and describes who needs to be contacted when and with what information. The Communication Plan also includes the plan for the deployment of personnel in all participating Project Teams and Core Teams.

**Risk Register:** aggregates the Risk Logs of all participating projects into a simple Excel format. The format of the Risk Log is equal to the one described above.

**Description of Rollout Window:** is usually accompanied by a visual presentation and summarizes major activities such a data migration or deployments on specific systems. An important item of this description is a table including all participating projects, their associated deployment times and points-of-use.

**One-Pager of all participating projects:** defines the scope of the rollout.

**Schedule Management:** a Rollout Schedule is quite similar to an ordinary project schedule; however, some distinct differences exist. IT-implementation projects usually have a timeframe of several months and are referred to ordinary projects in this paper. The release-specific rollout only lasts for a couple of hours. Ordinary project schedules are usually managed with duration-based or effort-based methods. A rollout can only be scheduled by duration-based approaches, since the duration of most activities is fixed. Here, the duration of tasks is usually scheduled on an hourly/minute basis.

Rollout Schedules need to have activity IDs and not only row numbers. What is the difference? Scheduling is commonly conducted with planning tools which facilitate row numbers. The planning tools use these line numbers for various calculations. However, they are also likely to be changed quite frequently if the plan is changed. In order to avoid communication issues, particularly during the operative rollout, alphanumerical IDs should additionally be used.

It is recommended to assign activity types to activities, see also Table 2. Why? Firstly, Rollout Schedules are very likely to be reused. This reusability is obviously much easier if unique activities, those that are just used once for a particular rollout, are sufficiently labelled. Additionally, a Rollout Plan is much easier to read if the operator knows already in advance, for example from the use of particular colours, what types of activities are expected, such as communication activities or checkpoints.

## Communication during release planning

Preparation work related to the release-specific rollout mainly covers coordination meetings with Project Teams and associated Core Teams participating in the release. Additionally, a large number of staff is required to be informed about the status related to the rollout preparation within the company. A rollout newsletter is a fit-for-purpose communication tool which can be easily customized to cover the coordination aspect and meet the corporate information need.

The release model illustrated here is used in such a manner that 6 rollouts are conducted on an annual basis, which results in an approximately 8-week preparation cycle for every rollout. A typical schedule related to the planning activities and publication dates of such a release cycle is illustrated in Fig. 6. The newsletter is published about every other week. It comprises topics related to the rollout preparation work, as well as subjects related to general IT questions such as new documents and procedures published by the security personnel, changes conducted by the Change Management etc. The newsletter is 2–6 pages long and announces deadlines and important dates, such as the date for the approval of

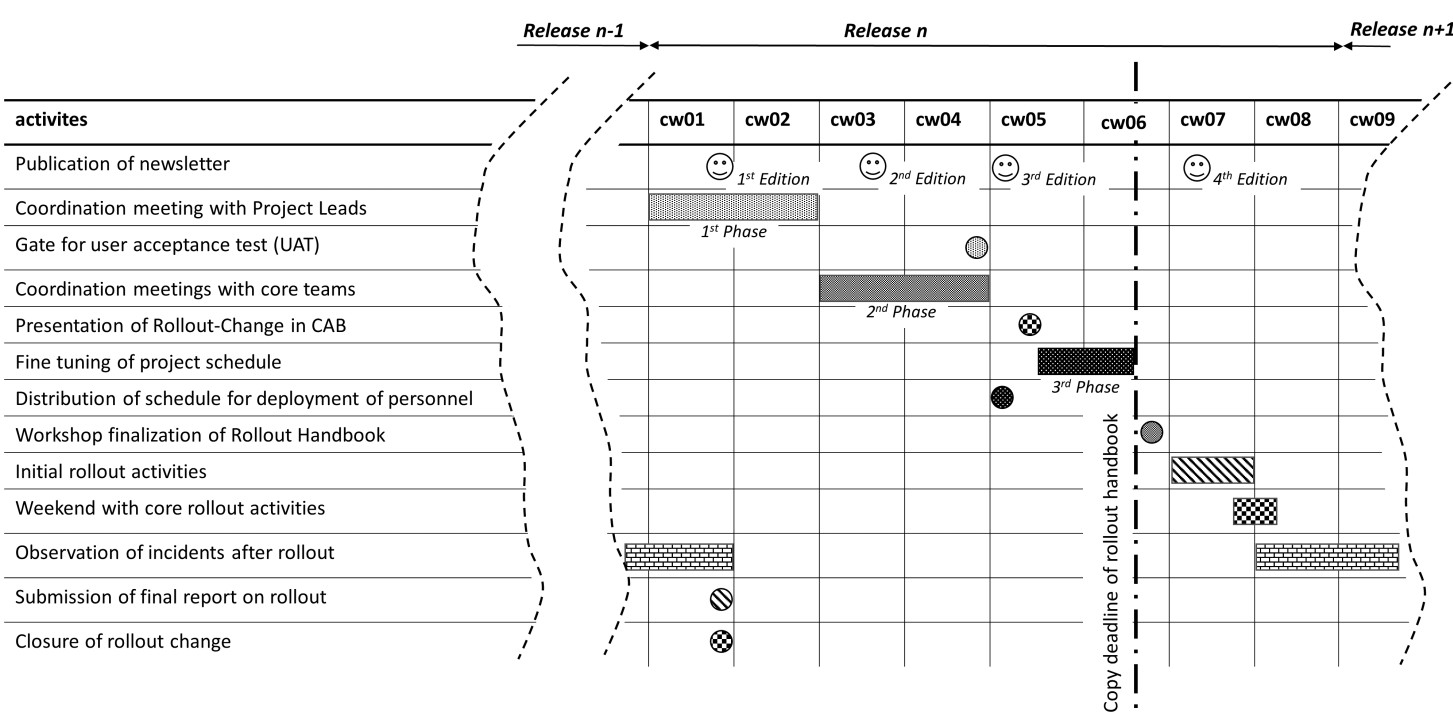

**Figure 6 Planning activities of the release.** Schedule of planning activities of the release, associated with the publication time of rollout newsletter.

the Rollout Handbook, which is done during a meeting with all key personnel. Documents related to the newsletter, such as the Rollout Handbook, are stored on MS SharePoint®. This provides the desirable function of enabling distribution of the newsletter to a wider community and limiting access to more confidential documentation.

The 4th edition is published a couple of days prior to rollout. By this stage the Rollout Handbook is in a final stage and everybody involved has a last chance to incorporate necessary changes. The newsletter publishes the major steps involved during the rollout, key contact phone numbers and an overview illustration. All interested parties thus know the overall picture of the rollout, whereas access to the Rollout Handbook is restricted. The 4th edition publishes all major planning steps and dates for the preparation work of the following rollout as well.

The 1st edition focuses on the incidents after the rollout. Incidents are observed up to 2 weeks after the rollout. The observation period closes with a Lessons-learned Workshop, the submission of the final report and the closure of the Rollout Change (see further below for the treatment of the Rollout Change).

The 2nd and 3rd editions provide temporary planning results and draft versions of the Rollout Handbook. Planning is conducted in three phases: during the 1st phase, interviews are conducted between Rollout Management and Project Teams, see Table 1 for the deliverable 'Dialogue with Rollout Management'. During the 2nd phase, Rollout Management interviews the Core Teams, and throughout the 3rd phase fine tuning of the Rollout Handbook is carried out.

## Rollout change

There is no universal measure on how to organize changes. Obviously, IT-service organizations need to record changes related to their system landscape; however, the way these are recorded, the level of detail and types of changes used differ significantly between organizations. What is a Rollout Change? The Rollout Change relates to the changes caused by the release-specific rollout. However, it only covers the release-specific Rollout Window. It has already been highlighted above that a project can cause damage in potentially two ways: firstly during the software deployment and secondly at the point-of-use. If the point-of-use falls outside the Rollout Window then it is not covered by the Rollout Change. In this case an additional change is required to be recorded. For this reason the Rollout Handbook contains a register which lists the deployment window and point-of-use for every project. The Rollout Handbook is appended to the Rollout Change as well.

The Rollout Change is created at the beginning of the release phase and then iteratively extended by updating the attached Rollout Handbook. Once this Handbook has been approved or is present in its final format, the change is requested to be approved by the Change Advisory Board (CAB). The Change is closed after the completion of the Lessons-learned Workshop, described in the next section.

## Lessons-learned workshop

Above it has already been explained that a Lessons-learned Workshop should be conducted after completion of the rollout. Lessons-learned Workshops provide many generic benefits, such as:

- the reuse of knowledge throughout an organization
- the reduction of project cost
- the improvement of methods and practices, and
- the enhancement of project delivery and quality management

just to mention a few of them. However, the Lessons-learned Workshop mentioned here relates solely to the rollout conducted and is not of generic nature. Let us step back a little to add some higher-level perspective.

Any IT organization has a need to improve its processes, ensure agreed service levels and justify its projects or initiatives. For this reason, Continuous Service Improvement (CSI) represents one of ITIL's five process groups. The CSI Register, which lists all improvement options and initiatives, is one of an IT organization's key deliverables (*Cabinet Office, 2011*). Key Performance Indicators (KPIs) are used to measure the process. How can we measure the success of a rollout? What do we mean by success? ITIL publications already suggest a variety of KPIs, such as the number of incidents caused after the release-specific rollout. These simple KPIs can, when implemented in practice, be quite tricky. Even though this particular KPI is suggested for the service validation and test process within ITIL's service transition process group (*Rance, 2011*), incidents are always a good indication of disturbances and are here also used for the evaluation of the rollout's success. Figure 7 illustrates the total number of incidents recorded within one IT

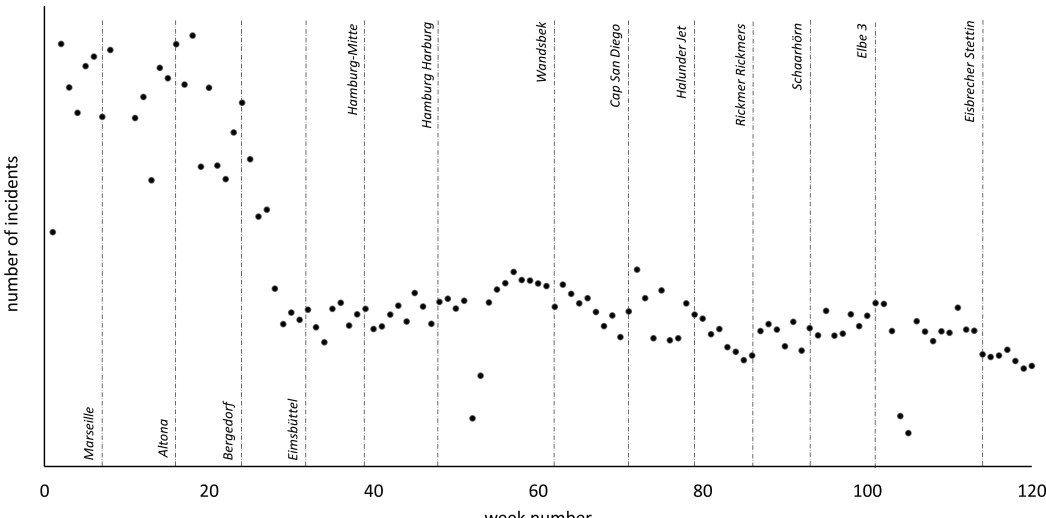

**Figure 7 Illustration of incidents during a two-year period.** Dotted lines illustrate week of rollout with release names; data relate to the total number of incidents recorded. Actual incident numbers are disclosed due to confidentiality reasons.

service organization. The dotted lines indicate the week during which a rollout occurred. The sharp decline of incidents during CW 30 is because a service-request process was introduced. The dips between 200 and 500 incidents/week are caused by the Christmas season, where many people are on holiday. Otherwise, it is quite difficult to acknowledge any impact of the rollout on the total number of incidents recorded. The picture changes once we take a closer look at a particular system. Figure 8 illustrates second-level incidents for a main order-handling system, which participated in all rollouts. These incidents account for about 5% of the total number of incidents recorded. After some rollouts a rise of incidents can be observed. How should we measure this?

Figure 9 shows the variability of incidents four weeks before and after rollout. The connecting lines are to aid visualisation of the form of the curves and are not intended to suggest data points at all. The data provided are normalized. Here, the mean value of incidents during a period of four weeks before rollout is set to 100%, with a standard deviation of ±18%. The figure suggests a significant increase of incidents 1–2 weeks after rollout. This increase is summarized in Fig. 10, where the average increase in incidents two weeks after rollout is plotted. Most rollouts show a significant increase. The negative value occurred during the summer holiday period time, when less people recorded and reported incidents.

The values provided have been incorporated for illustrative purposes. It is hardly surprising that if the number of projects participating in a rollout increases, then the number of incidents increases as well. Figure 11 shows a correlation coefficient of 0.4 between the average increase of incidents two weeks after rollout and the number of participating projects per release. If we consider the complexity of a rollout as introduced above, see also Fig. 5, which is the total number of interactions between Project Teams and Core Teams participating in a rollout, then we obtain a similar correlation coefficient of

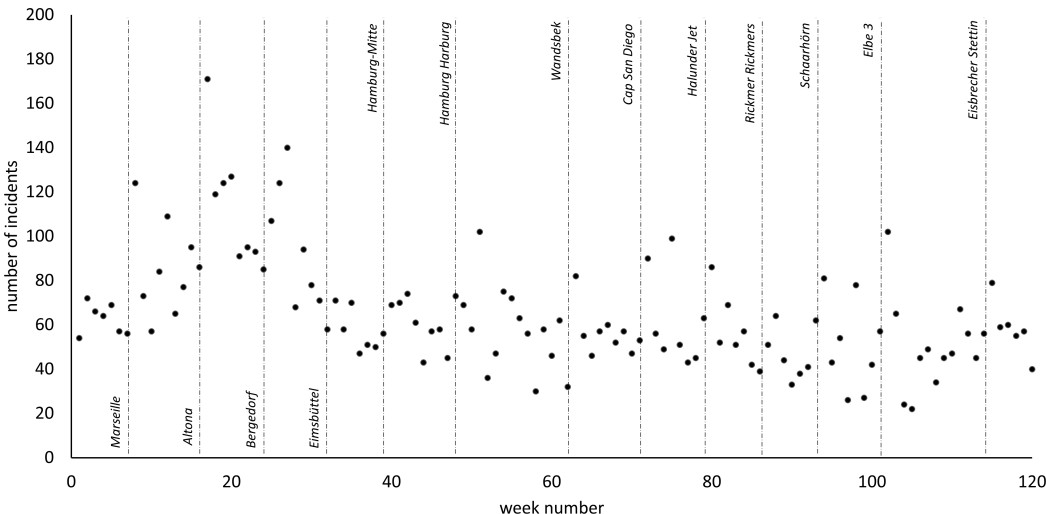

**Figure 8 Second-level incidents of a main order-handling system.** Illustration of incidents during a two-year period. Dotted lines illustrate week of rollout with release names; data relate to second-level incidents of a main order-handling system, which participates in all rollouts.

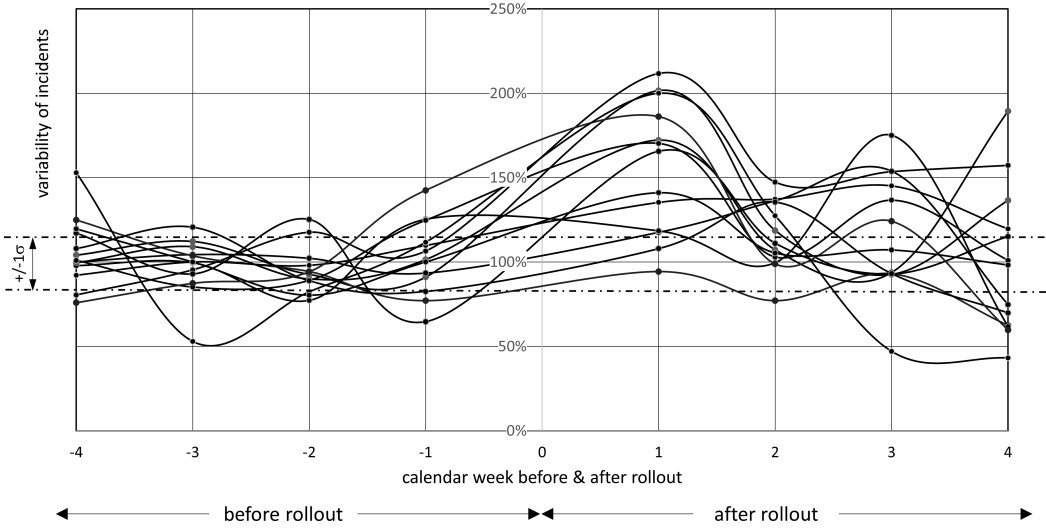

**Figure 9 Variability of incidents four weeks before and after rollout.** Connecting lines visualize the form of the curves; the mean value of incidents during a period of four weeks before rollout is set to 100%, with a standard deviation of ±18%.

0.36, see Fig. 12. The high level of incidents provides a good primary source of information on how well the rollout and release performed. However, these data should only be used indicatively and need to be interpreted carefully, as:

- processes or the ways processes are managed change over time, such as the above-mentioned introduction of a service-request process, which reduced the number of incidents

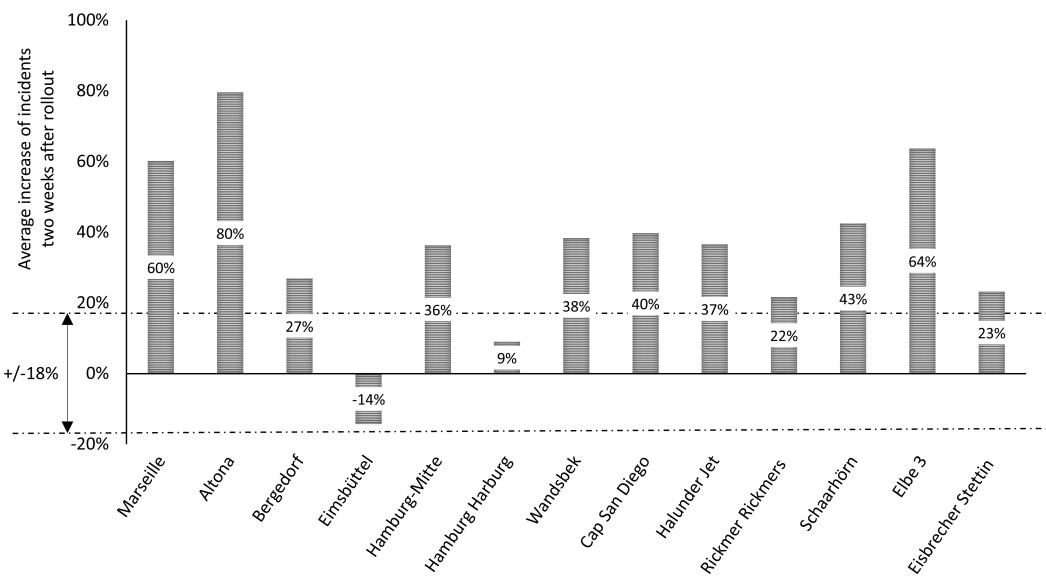

**Figure 10 Average increase of incidents two weeks after rollout.** Standard deviation is ±18% as illustrated in Fig. 9; names label the rollouts, which are plotted on a timeline in Fig. 8.

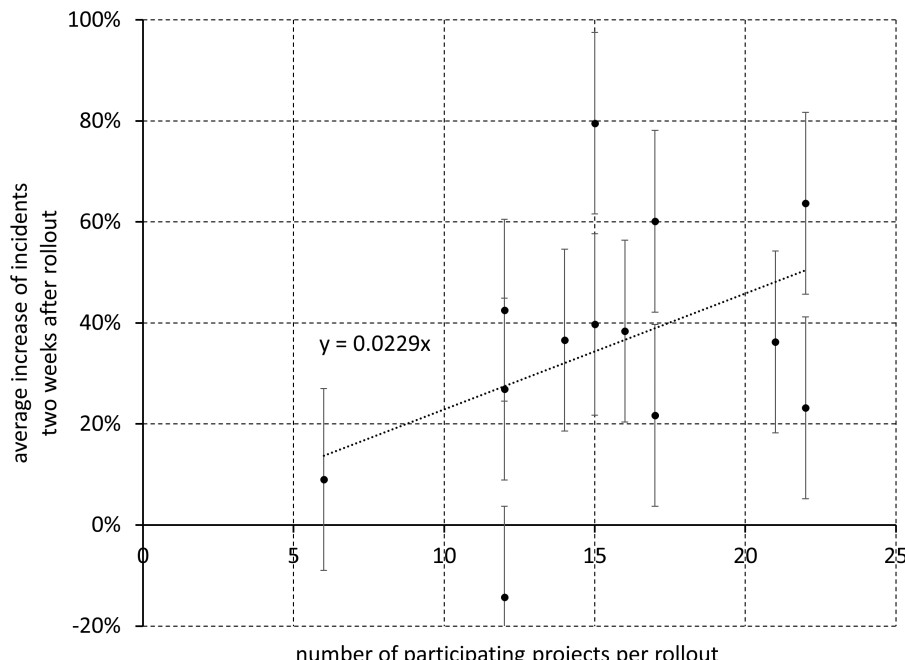

**Figure 11 Correlation coefficient A.** Correlation coefficient of 0.4 between the average increase in incidents two weeks after rollout and the number of participating projects per release.

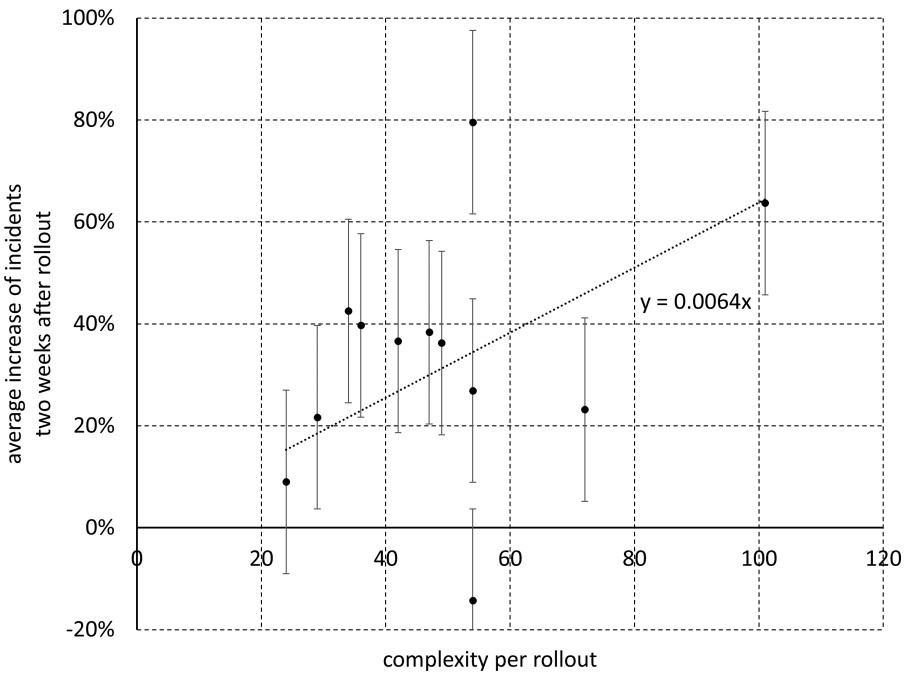

**Figure 12 Correlation coefficient B.** Correlation coefficient of 0.36 between average increase of incidents two weeks after rollout and the complexity per rollout.

- some projects separate their cut-over and point-of-use. Incidents can, for example, pop up two or three weeks after rollout once the software goes live
- the number of incidents recorded is impacted by seasonal variability, such as holiday periods and Christmas
- changes are conducted after and prior to rollout, and can significantly influence the statistics.

In order to learn what actually caused these incidents they need to be examined in a more detailed manner, which is the purpose of the Lessons-learned Workshop recommended here. The Lessons-learned Workshop should be attended by all key personnel involved, such as the Project Manager/Cut-over Manager of the participating projects, Test Manager, and staff involved in the resolution of 2nd and 3rd level incidents—just to mention some of them. In order to run such a workshop it is essential to identify the incidents/service requests most frequently caused by the rollout and to classify them. These incident categories are used solely for rollout-evaluation purposes and additionally to the ordinary classification scheme, which is part of the incident-management process.

Table 3 provides an example of typical incident categories used after rollout evaluation. Between 60 and 80% of the incidents evaluated have usually not been detected by the test. Incidents caused due to bad Rollout Schedule or descriptions are rare. This suggests that the evaluation of incidents as such is more a measure of the test quality than of the Rollout Planning and execution performance.

**Table 3  Incident categories used solely for rollout evaluation.** This classification scheme isused in addition to the schema already facilitated by the incident management process, which is not further described here.

| Incident category | Description |
| --- | --- |
| Error within release/ not detected through testing | Incident was caused by a software error which was not detected in testing. This category is also used for service requests, which fix release-specific errors, e.g., in the form of patches. |
| Human failure | This category is used for incidents caused by human failure, such as faulty execution of patches or noncompliance with agreements. |
| Incident caused by project | Incident is related to a project; this category makes particular sense if a significant number of incidents are related to a particular project. |
| Rollout Schedule requires change | Activity required to be conducted was not properly described. If it is a reoccurring activity then the description needs to change. |
| Expected behaviour | Irregularities recorded are considered to be an accepted and expected behaviour of the system or application. |
| Sequential error within the Rollout Schedule (application-specific) | Incident is caused by a sequential error within the Rollout Schedule; activity was requested to be conducted by an application which regularly participates during the rollout. |
| Sequential error within the Rollout Schedule (project-specific) | Incident is caused by a sequential error within the Rollout Schedule; the activity was requested to be conducted by a Project Team. |
| Bugfix | This category relates to incident-response measures which occur prior to the rollout and which aim to eliminate software error. It is merely a convention to relate them to the Rollout Change, which also fixes these errors. |
| Missing contact person | The contact person who logged this change could not be contacted. The number of incidents assigned to this category should be kept small. |
| Incident is not reproducible | The incident cannot be reproduced. |
| Wrongly connected | Incident was not related to the Rollout Change and was wrongly related. |
| Not classifiable | Incident cannot be classified with available classification; the classification is of course extended with additional categories where required. |

One of the challenges these workshops face is related to the proper assignment of incidents to the Rollout Change. The application-specific increase of 2nd level incidents after rollout provides a good indication of the expected number of incidents required for analysis.

The second KPI introduced is specifically designed to measure the planning and execution performance of a rollout. Here, the scheduled time is compared with the actual execution time. The measurement can easily be conducted by the inclusion of communication items within the Rollout Schedule, such as e-mails. These e-mails are sent by the rollout personnel shortly prior to execution of critical items, here called 'measurement points'. The time stamps of these e-mails provide a measure of the actual execution time. Figure 13 illustrates the discrepancy between the scheduled time (open circles) and actual execution time (closed circles). If the closed circles are below the open ones then the actual rollout was executed quicker than foreseen. If vice versa, a delay occurred. Figure 13 illustrates that the activity related to measurement point 2 finished sooner than foreseen. The activity related to measurement point 3, however, was delayed.

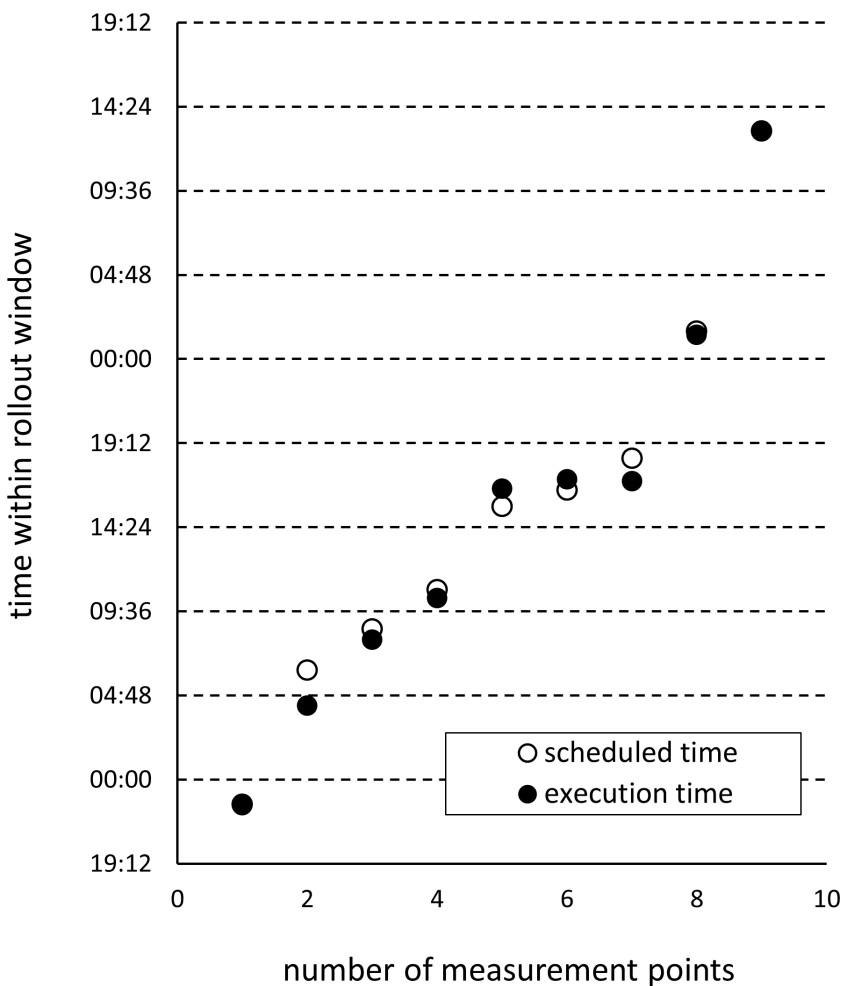

**Figure 13 Discrepancy between the scheduled time and actually executed time.** Illustration of the discrepancy between the scheduled time and actually executed time for nine measurement points.

Such a diagram provides a good basis for a structured discussion on why certain rollout activities started later or finished earlier.

## CONCLUSIONS

Within this publication a deployment framework is presented which illustrates how good ITIL management practices can be aligned with a project-management model in order to assure timely delivery of what this paper refers to as 'project-specific cut-over'. This alignment is shown by the use of a semi-agile release model. Due to the release model selected, which is particularly characterized by its bundling of projects for what is here referred to as 'release-specific rollout', a new definition and interpretation of ITIL's generic view of deployment is required. The facilitated release model additionally required two types of go-live scenarios: one for each participating project and one for each release.

In order to finalize a project-specific cut-over punctually, a variety of cut-over related deliverables are presented and chronologically assigned to a typical IT project managed in a

cascading manner. Projects already need to take into account cut-over related deliverables during the initial planning phase. It is also shown how project-specific outcomes interrelate with release-specific planning activities and are finally aggregated within the Rollout Handbook and proposed as Rollout Change for approval by the CAB. Emphasis is placed on the Lessons-learned Workshop recommended to be conducted a couple of weeks after the rollout, once the level of incidents has normalised. This Workshop should be guided by the rollout/cut-over related incidents, and a comparison of the scheduled time and actual execution time of the rollout should be included in the Workshop.

Even though consulting methodologies such as ASAP (Accelerated SAP) recommend scattered, project-specific deliverables useful for Cut-over Planning, this publication offers an integrated approach on how to systematically prepare for a project-specific cut-over with all required deliverables. The framework provided extends the ITIL release and deployment process by means of IT projects and allows IT projects to interface easily with the ITIL change-management process.

The framework has proven to be successful since it forces projects to think already during the planning phase about cut-over related topics. After implementation of this framework we see projects better prepared prior entering the release. However, we also do have projects which wish to cut-over by the use of their own methodology. Probably this formal approach is not everybody's cup of tea.

## ACKNOWLEDGEMENTS

I would especially like to thank Andreas Veldmann for the constantly inspiring discussions and his substantial help in the implementation of this framework. Thanks are also due to Necla Demirci, Gerd Heyken and Joachim Kunze for many stimulating discussions on this paper and related work. I am particularly grateful to Sebastian Evert, Dr Sebastian Diedrich and Enrico Voigt for providing incident-related data from Cherwell's IT Service Management Software. I am very grateful to Jim Blake at World2World Hamburg for editing the English version of this paper.

### Funding

The research related to the implementation of the framework described in this paper has been made possible by the Department of Test and Release Management of Otto (GmbH & Co KG) which is a member of the otto group. The framework presented here in the form of an illustrative case study arose during my consultancy work and has been implemented for some applications within the Department of Testing and Release Management at the Otto (GmbH & Co KG).

### Grant Disclosures

The following grant information was disclosed by the author:
Department of Test and Release Management of Otto (GmbH & Co KG).

## Competing Interests

Dr. Guido Nageldinger is an employee of Otto (GmbH & Co KG) which is a member of the otto group.

## Author Contributions

- Guido Nageldinger conceived and designed the experiments, performed the experiments, analyzed the data, wrote the paper, prepared figures and/or tables, performed the computation work.

## Supplemental Information

Supplemental information for this article can be found online at http://dx.doi.org/10.7717/peerj-cs.29#supplemental-information.

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
