# Peer review of "A framework for cut-over management"

_PeerJ Computer Science, doi:10.7717/peerj-cs.29_

## Round 0.1 · original submission · Minor Revisions

This is a very nice paper. However, I believe that additional research references, related to the area addressed by the paper, are needed. I understand there are not many people working in the area. Perhaps 2 or 3 papers that are obliquely related to the area could be possible. That is, that this is not a direct extension of their research, but that they work in related areas, and how the work in this paper is different.

Other than that, however, the paper is well written and both reviewers liked it.

Reviewer 1 ·

Basic reporting

This paper provides a governance structure for IT-related projects, especially when the projects are ready to roll out to production. This paper is based on the author’s consulting experiences in the Department of Testing and Release Management at the Otto Group, which is headquartered in Hamburg, Germany.

My impression of the paper is that the author has done enough work to warrant a publication, especially the author does provide a methodology for rolling out projects so that the transition to new systems can be done smoothly.

However, I do have one complaint. The paper does not include a real-world case. (Maybe I missed it, I do not know.) For papers of this sort, real-world cases are important because the theory proposed by the paper can only be tested empirically, not mathematically. Hence, a successful application of the methodology is important to convince readers that the methodology is valid.

Experimental design

See previous comments.

Validity of the findings

Can only be validated by real-world cases.

Additional comments

Please provide a real-world case of successful application of your methodology.

·

Basic reporting

The paper clearly defines a method for live deployment of a project. It does a good job of emphasizing the need for applying an exact, formal methodology as opposed to simply using best practices, so that the scientific method can be applied and and data can be used to back up our idea that the method is "better".

Experimental design

The paper demonstrates an adequate case study that applies this methodology and the ideas are well supported. Because each project is unique, it would be difficult or impossible to exactly reproduce the results. However, it could be tested on other projects to see if inclusion of cut-over requirements in the ITIL management model show more timely delivery.

Validity of the findings

The paper used good statistical methods that support the validity of the papers findings

Additional comments

No Comment

---

## Round 0.2 · accepted · Accept

Good paper thank you for the revision